# The Gastroprotective Effects of *Anisomeles indica* against Ethanol-Induced Gastric Ulcer through the Induction of IκB-α and the Inhibition of NF-κB Expression

**DOI:** 10.3390/nu16142297

**Published:** 2024-07-17

**Authors:** Yu-Ru Chen, Hsiu-Man Lien, Fuu-Jen Tsai, Jiunn-Wang Liao, Yng-Tay Chen

**Affiliations:** 1Graduate Institute of Food Safety, College of Agriculture and Natural Resources, National Chung Hsing University, Taichung 402202, Taiwan; angus02005@gmail.com; 2Research Institute of Biotechnology, Hungkuang University, Taichung 433304, Taiwan; herb736@sunrise.hk.edu.tw; 3Genetic Center, Department of Medical Research, China Medical University Hospital, Taichung 404328, Taiwan; 000704@tool.caaumed.org.tw; 4School of Chinese Medicine, China Medical University, Taichung 404328, Taiwan; 5Graduate Institute of Pathobiology Veterinary, National Chung Hsing University, Taichung 402202, Taiwan; jwliao@dragon.nchu.edu.tw; 6Department of Food Science and Biotechnology, College of Agriculture and Natural Resources, National Chung Hsing University, Taichung 402202, Taiwan; 7Department of Post-Baccalaureate Medicine, College of Medicine, National Chung Hsing University, Taichung 402202, Taiwan

**Keywords:** *A. indica*, gastric ulcer, NF-κB, IκB-α

## Abstract

*Anisomeles indica* (L.) Kuntze is a traditional herb with multiple medicinal properties and with potential for preventing or treating various diseases. Acteoside, one of the active ingredients in *A. indica*, is prepared into commercially available products of A. indica HP813 powder. In this study, the gastroprotective effects of *A. indica* HP813 powder were evaluated. Wistar rats were treated with *A. indica* HP813 powder at doses of 0, 207.5, 415, and 830 mg/kg body weight for 28 days. Then, gastric ulcers were induced by the oral administration of 70% ethanol (10 mL/kg body weight) on day 28. The rats were sacrificed at the end of the trial, and stomach tissues were collected. These stomach tissues were then used for macroscopic, microscopic, and immunohistochemical analyses. The results indicated that the area of gastric ulcer was 48.61%, 35.30%, and 27.16% in the ethanol-induced group, 415 mg/kg *A. indica* HP813 powder group, and 830 mg/kg *A. indica* HP813 powder group, respectively. In addition, the lesion scores were 2.9, 2.4, and 2.3 in the ethanol-induced group, 415 mg/kg *A. indica* HP813 powder group, and 830 mg/kg *A. indica* HP813 powder group, respectively. The immunochemical staining of the gastric tissue revealed that *A. indica* HP813 powder reduced the expressions of TNF-α and NF-κB proteins in the gastric tissue, which had been induced by ethanol. Finally, *A. indica* HP813 powder protected the gastric ulcer from ethanol damage through IκB-α induction. The present results demonstrated that *A. indica* HP813 powder has protective effects against ethanol-induced gastric ulcer.

## 1. Introduction

Gastric ulcer is a common digestive health condition. The causes of gastric ulcer include excessive alcohol consumption, smoking, stress, excessive gastric acid secretion, *Helicobacter pylori* infection, and side effects of medication [1,2,3]. It is believed that the pathogenesis of gastric ulcer is related to inflammation and that the aforementioned causes can lead to inflammation [4].

Alcohol intake is strongly associated with stomach problems, including gastritis, gastric ulcer, and even stomach cancer [5,6]. Alcohol triggers an acute inflammatory response and stimulates the immune system, altering the levels of proinflammatory cytokines; this process further triggers apoptosis [3,7]. Therefore, an excessive intake of alcohol increases the levels of inflammatory cytokines and oxidative stress, leading to mucosal cell destruction, gastric epithelial cell necrosis, and inflammatory cell infiltration and resulting in gastric mucosal injury, inflammation, and ulceration [8]. Regarding inflammatory cell infiltration, neutrophils accumulate in the lesion, increasing the activity of myeloperoxidase (MPO) in these neutrophils. MPO then produces hypochlorous acid, which causes acute inflammation and gastric mucosal damage [9]. Previous studies have indicated associations between ethanol-induced gastric ulcer elevations in inflammatory cytokines, such as interleukin-1β (IL-1β), interleukin-6 (IL-6), heme oxygenase-1 (HO-1), and tumor necrosis factor-α (TNF-α) [10].

*Anisomeles indica* (L.) Kuntze, a plant of the genus Anisomeles in the family Lamiaceae [11,12], is a traditional herbal medicine commonly used in Taiwan. The results of recent studies have revealed that *A. indica* extracts have pharmacological activities, including anti-inflammatory activities, tumor cell growth inhibition, treatment of gastrointestinal and liver diseases, treatment of inflammatory skin diseases, and anti-H. pylori activities [13,14,15]. Previous studies have isolated some chemical constituents from *A. indica*, including apigenin, ovatodiolide, β-sitosterol, and acteoside [16,17]. Acteoside is one of the active chemical constituents of the *A. indica* plant, which has exhibited anti-inflammatory and anticancer activities [18,19,20]. TNF-α is an important mediator for the occurrence of gastric ulcers because it hinders the blood microcirculation around the ulcer and activates NF-κB, thereby regulating the expression of various inflammatory mediators [21]. Activated IκB undergoes ubiquitylation and proteasomal degradation, and the freed NF-κB is translocated to the nucleus; this step is critical in the acceleration of the inflammatory response [21]. The objective of this study is to determine whether *A. indica* HP813 powder inhibits ethanol-induced gastric ulcers and whether it can reduce the inflammatory response of gastric ulcers caused by ethanol.

## 2. Materials and Methods

### 2.1. Sample

*A. indica* HP813 powder was provided by SYI Biotechnology Co., Ltd. (Taichung, Taiwan). The whole *A. indica* plant was obtained from the contractual farms of SYI Biotechnology (Taichung, Taiwan). *A. indica* extract was prepared as previously described [15] with minor modifications. The formulation of *A. indica* HP813 powder included soluble fiber from corn, xylo-oligosaccharides, and *A. indica* extract powder. The functional materials were xylo-oligosaccharides and acteoside. This powder was soluble in water. The main efficacy ingredient of *A. indica* HP813 powder is acteoside. The content of acteoside in *A. indica* HP813 powder (2000 mg) is 55 μg. In this study, the content of acteoside in the low-dose group was 27.5 μg, the medium-dose group was 55 μg, and the high-dose group was 110 μg.

### 2.2. Animals

Five-week-old male Wistar rats were purchased from BioLASCO Taiwan Co., Ltd. (Taipei, Taiwan) and were housed at the Research Center for Animal Medicine, National Chung Hsing University. The animals’ housing conditions included a temperature of 22 °C ± 1 °C and a 12 h light–dark cycle. These animals were used experimentally in accordance with the *Guidelines for the Care and Use of Laboratory Animals* published by the Chinese Society for the Laboratory Animal Science. This study was approved by the Institutional Animal Care and Use Committee of National Chung Hsing University (IACUC No.: 111-043).

### 2.3. Animal Experimental Design

Forty rats were randomly divided into five groups, with eight rats in each group. The experiment was conducted over 28 days (4 weeks), with oral gavage provided every day. The gavage dose was provided on the basis of body weight (10 mL/kg body weight) and was changed weekly. The recommended daily intake of *A. indica* HP813 powder for an adult weighing 60 kg is 4000 mg, which is 67 mg/kg/day. In terms of experimental animals, multiplying by 6.2 equals the recommended dose of 415 mg/kg/day, 830 mg/kg is twice the recommended dose, and 207.5 mg/kg is half the recommended dose. The groups are described as follows (Figure 1):

Group 1: The control group received a daily administration of distilled water by oral gavage for 28 days.

Group 2: The positive control group received a daily administration of distilled water by oral gavage for 28 days, and on day 28, this group received a single dose of 70% EtOH (10 mL/kg body weight) by oral gavage.

Group 3: The low-dose group received a daily administration of *A. indica* HP813 powder (207.5 mg/kg body weight) by oral gavage for 28 days, and on day 28, this group received a single dose of 70% EtOH (10 mL/kg body weight) by oral gavage after 1 h of *A. indica* HP813 powder treatment.

Group 4: The medium-dose group received a daily administration of *A. indica* HP813 powder (415 mg/kg body weight) by oral gavage for 28 days, and on day 28, this group received a single dose of 70% EtOH (10 mL/kg body weight) by oral gavage after 1 h of *A. indica* HP813 powder treatment.

Group 5: The high-dose group received a daily administration of *A. indica* HP813 powder (830 mg/kg body weight) by oral gavage for 28 days, and on day 28, this group received a single dose of 70% EtOH (10 mL/kg body weight) by oral gavage after 1 h of *A. indica* HP813 powder treatment.

Prior to sacrifice, all animals were deprived of food and water overnight. On day 29, the rats were anesthetized with 2% isoflurane and were sacrificed by blood sampling through the celiac artery. After the rats were sacrificed, the cardia and pylorus were tied with a thin thread, the ends of the esophagus and duodenum were cut separately, and the stomach was dissected. After the stomach was removed, a cut was made along the great curve, the stomach was washed with phosphate-buffered saline, and images were captured. These images were then used for the quantitative analysis of the gastric ulcer lesion area. Finally, tissues were removed and fixed in 10% neutral formalin for histopathological examination.

### 2.4. Histopathological Study

The vital organs and stomachs of the rats in all the experimental groups were fixed with 10% neutral formalin for 1 week and then trimmed, dehydrated, cleaned, infiltrated in paraffin, and embedded. Subsequently, 3 μm thick tissue sections were cut with a tissue slicer (Leica RM 2245, Nussloch, Germany) and stained with hematoxylin and eosin; histopathological changes were then observed under an optical microscope. Regarding the evaluation criteria for histopathological changes, the degree of lesions was divided into 5 grades: grades 1, 2, 3, 4, and 5 represented minimal lesions (<1%), slight lesions (1–25%), moderate lesions (26–50%), moderately severe lesions (51–75%), and extremely severe lesions (76–100%), respectively [22].

### 2.5. Quantitative Analysis of Gastric Ulcer Lesion Area

The captured images of the gastric tissue were quantitatively analyzed using ImageJ software (Version 1.53k, National Institutes of Health, Bethesda, MD, USA, 6 July 2021). The gastric lesions were framed and quantified as the total area of the stomach, and a color threshold was employed to select and quantify the lesions. The percentage of the damaged area of the gastric mucosa in rats with alcohol-induced acute gastric ulcer was calculated using the following formula: (area of gastric injury/total area of stomach) × 100%.

### 2.6. Immunohistochemistry

The immunohistochemistry method is described briefly as follows. For paraffin tissue slice deparaffinization and rehydration, the slices were incubated in 3% H_2_O_2_ solution for 30 min. For antigen retrieval, the slices were boiled in 0.01 M citrate buffer for 20 min and then washed in 50 mM Tris-HCl (pH 7.6) with 0.05% Tween for 2 min. The slices were then incubated with 5% nonfat dry milk for 30 min to block nonspecific binding. The slices were then hybridized with anti-TNF-α antibodies (1:200; GeneTex, Inc., Hsinchu, Taiwan), anti-NF-κB antibodies (1:200; GeneTex, Inc., Hsinchu, Taiwan), and anti-IκB-α antibodies (1:200; GeneTex, Inc., Hsinchu, Taiwan) for 1 h before being incubated with a secondary antibody. Finally, the slices were incubated with a peroxidase-labeled streptavidin–biotin complex and diaminobenzidine substrate to identify TNF-α-, NF-κB-, and IκB-α-labeled cells. The immunohistochemistry evaluation was determined semi-quantitatively in 50 cells of each sample using the following score for cytosol staining: 1—no, 2—weak, 3—moderate, and 4—strong. The average intensity of the immune reaction has to be given as the number of cells of each type x corresponding coefficient (1, 2, 3, or 4)/total number of cells.

### 2.7. Statistical Analysis

Statistical analyses of the data between two groups were determined by using post hoc *t*-tests. Data were shown as means ± standard deviations. A *p* value of <0.05 indicated statistical significance. 

## 3. Results

### 3.1. A. indica HP813 Powder Protection against Ethanol-Induced Gastric Ulcers

The gastric mucosa of the control group was intact without injury (Figure 2A), whereas the positive control group exhibited evident bleeding and gastric ulcers (Figure 2B). The condition of the gastric ulcers in the 207.5 mg/kg group was not significantly improved (Figure 2C). The gastric mucosa in the 430 mg/kg and 830 mg/kg groups was damaged, but this damage was more minor than that in the positive control group (Figure 2D,E).

### 3.2. A. indica HP813 Powder Decreased the Area of Gastric Ulcers Caused by Ethanol

The average percentages of gastric mucosa injury were 48.61% ± 16.24% in the positive control group, 45.22% ± 13.06% in the 207.5 mg/kg group, 35.30% ± 6.97% in the 415 mg/kg group, and 27.16% ± 8.89% in the 830 mg/kg group (Table 1). Compared with the positive control group, the percentages of gastric mucosa injury were significantly decreased in the 415 mg/kg group and the 830 mg/kg group (*p* < 0.05). No significant differences were found between the 207.5 mg/kg group and the positive control group (*p* > 0.05).

### 3.3. A. indica HP813 Powder Decreased the Histopathological Lesion Score

The total average pathological scores for gastric injury—including degeneration/necrosis, hemorrhage, inflammatory response, and edema (Figure 3)—in the control, positive control, 207.5 mg/kg, 415 mg/kg, and 830 mg/kg body weight groups were 0, 2.9 ± 0.9, 2.9 ± 0.8, 2.4 ± 0.7, and 2.3 ± 0.8, respectively (Table 2). The subtotal mean score of gastric injury was significantly increased in the positive control group compared with the control group (*p* < 0.05), and the mean score was significantly decreased in the 430 and 830 mg/kg body weight groups compared with the positive control group (*p* < 0.05). Appendix A showed gastric injury—including degeneration/necrosis, hemorrhage, inflammatory response, and edema. The normal control and positive control are shown in Appendix A; the low-dose group (207.5 mg/kg), medium-dose group (415 mg/kg), and high-dose group (830 mg/kg) are shown in Appendix A.

### 3.4. A. indica HP813 Powder Decreased Ethanol-Induced TNF-α and NF-κB Expression and Increased IκB-α Expression

Ethanol administration significantly increased TNF-α protein expression in the rats’ gastric mucosa tissue compared with the control group (Figure 4A,B). Moreover, TNF-α protein expression significantly decreased in the 207.5, 415, and 830 mg/kg body weight groups compared with the positive control group (Figure 4C–E). The average intensities of TNF-α expression by semi-quantitative IHC were 1.14 ± 0.03, 2.3 ± 0.03, 2.0 ± 0.09, 1.74 ± 0.06, and 1.35 ± 0.07, respectively (Figure 4F, Appendix A). Ethanol administration significantly increased NF-κB protein expression in rats’ gastric mucosa tissue compared with the control group (Figure 5A,B). NF-κB protein expression significantly decreased in the 207.5, 415, and 830 mg/kg body weight groups compared with the positive control group (Figure 5C–E). The average intensities of NF-κB expression by semi-quantitative IHC were 2.27 ± 0.01, 3.29 ± 0.04, 3.02 ± 0.07, 2.64 ± 0.07, and 2.48 ± 0.02, respectively (Figure 5F, Appendix A). Ethanol administration significantly decreased IκB-α protein expression in the rats’ gastric mucosa tissue compared with the control group (Figure 6A,B). In addition, IκB-α protein expression significantly increased in the 207.5, 415, and 830 mg/kg body weight groups compared with the positive control group (Figure 6C–E). The average intensities of IκB-α expression by semi-quantitative IHC were 3.26 ± 0.06, 2.17 ± 0.02, 2.78 ± 0.04, 3.07 ± 0.19, and 3.27 ± 0.05, respectively (Figure 6F, Appendix A).

## 4. Discussion

The present study administered a single oral dose of 70% EtOH to induce gastric ulcers in Wistar rats. The area of gastric mucosal damage in the positive control group was 48%, indicating that 70% EtOH effectively induced gastric ulcers. Several methods—including the oral gavage of alcohol or nonsteroidal anti-inflammatory drugs (NSAIDs) [23,24], cold-restraint stress [25], and pyloric ligation [26]—have been used to establish animal models of gastric ulcers [27]. Common clinical treatments for gastric ulcers, such as proton pump inhibitors [28,29], can regulate gastric acid secretion and prevent the side effects of NSAIDs [30]. However, the prolonged use of proton pump inhibitors can cause side effects such as kidney damage, hypomagnesemia, and bone fractures [31,32]. Accordingly, research is beginning to focus on identifying which herbs can effectively treat gastric ulcers to understand which active ingredients are effective in this regard [26].

Bleeding, inflammation, and apoptosis are the main pathogenic mechanisms of gastric ulcers. Accordingly, to delay the progression of or treat gastric ulcers, bleeding, inflammation, and apoptosis must be controlled. In the present study, the gastric mucosa gross findings revealed a reduction in the area of gastric mucosal damage in the medium- and high-dose groups compared with the positive control group. These findings indicate that medium to high doses of *A. indica* HP813 powder can mitigate inflammation and bleeding; this indication was further confirmed by quantitative analysis. Apoptosis in the gastric mucosa is characterized by tissue degeneration and necrosis. The tissue lesion assessment revealed that the scores of degeneration/necrosis in the medium- and high-dose groups were significantly lower than those in the positive control group, indicating that *A. indica* HP813 powder mitigated apoptosis caused by gastric ulcers. 

Our results demonstrated that ethanol-induced gastric mucosa injury caused inflammation, which significantly increased the average intensity of TNF-α and NF-κB in the stomach tissues in the untreated group by semi-quantitative IHC analysis. NF-κB expression was activated, leading to an increase in the production of inflammatory cytokines. Consequently, NF-κB was translocated to the nucleus, increasing the transcription of inflammatory cytokines such as TNF-α, IL-1β, and IL-6 [33,34,35]. NF-κB is a master regulator in the transduction cascade of inflammatory responses; it operates mainly through the stimulation of proinflammatory cytokines such as TNF-α [33,35,36]. In the inactive condition, NF-κB binds to its inhibitory protein (IκB). When stimulated, IKK induces IκB phosphorylation, resulting in IκB degradation [21,37]. Subsequently, NF-κB is released and translocated to the nucleus, culminating in the transcription of genes encoding proinflammatory cytokines [38]. Thus, to identify effective treatments for ethanol-induced gastric ulcers, therapeutic approaches targeting the activation of the NF-κB pathway should be investigated. In the present study, *A. indica* HP813 powder decreased TNF-α-induced IκB-α degradation and NF-κB average intensity mediated by ethanol-induced gastric ulcers. Accordingly, we propose that *A. indica* HP813 powder inactivates NF-κB signaling through its anti-inflammatory properties.

In this study, the main efficacy ingredient of *A. indica* HP813 powder is acteoside, and the gastroprotective effect in medium-dose *A. indica* Hp813 powder content was 55 μg of acteoside. Previous studies showed acteoside (40 mg/kg) anti-ulcerogenic activity through the neutralization of acid and gastric secretion by inhibiting H+/K+-ATPase [39] and related to antioxidant, neuroprotective, anti-inflammatory, and cytoprotective activities [40]. Future studies are needed to research the mechanisms of gastroprotective function by *A. indica* HP813 powder.

## 5. Conclusions

In the present study, acteoside, an ingredient in *A. indica* extract, was formulated as *A. indica* HP813 powder for the assessment of gastrointestinal functional improvement. An in vivo study revealed that doses of *A. indica* HP813 powder at 415 and 830 mg/kg body weight were effective for ameliorating 70% EtOH-induced acute gastric ulcers. The results indicated that the proposed *A. indica* extract could be used in potential products for the treatment of gastrointestinal health conditions.

## Figures and Tables

**Figure 1 nutrients-16-02297-f001:**
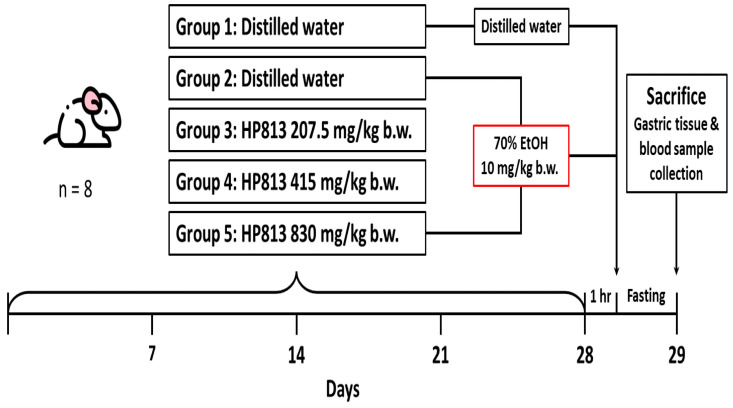
Animal experimental procedure for the evaluation of gastrointestinal functional improvement due to *A. indica* HP813 powder in the 70% EtOH-induced gastric ulcer rat model.

**Figure 2 nutrients-16-02297-f002:**
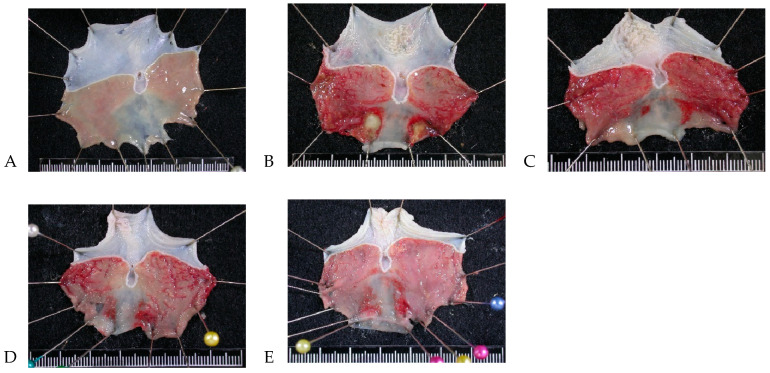
Gastric mucosa gross findings regarding protective effects of *A. indica* HP813 powder in rats. (**A**) Control group, (**B**) positive control group, (**C**) 207.5 mg/kg body weight group, (**D**) 415 mg/kg body weight group, and (**E**) 830 mg/kg body weight group.

**Figure 3 nutrients-16-02297-f003:**
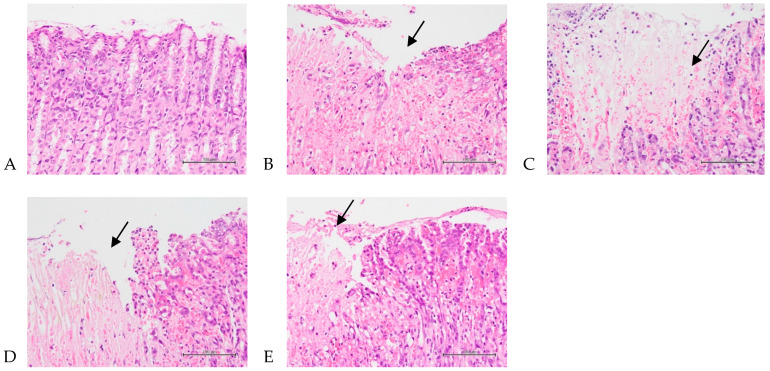
Histopathological changes in the stomach representing the protective effects of *A. indica* HP813 powder on the gastric mucosa in rats. (**A**) No significant change in the control group. (**B**) Seventy percent EtOH caused multifocal slight to moderate/severe degeneration/necrosis with hemorrhage, inflammation, and edema in the gastric mucosa in the positive control group (arrow points to the injury). (**C**–**E**) Seventy percent EtOH caused multifocal minimal to moderate/severe degeneration/necrosis with hemorrhage, inflammation, and edema in the gastric mucosa in the 207.5, 415, and 830 mg/kg body weight groups (arrow points to the injury, 400×).

**Figure 4 nutrients-16-02297-f004:**
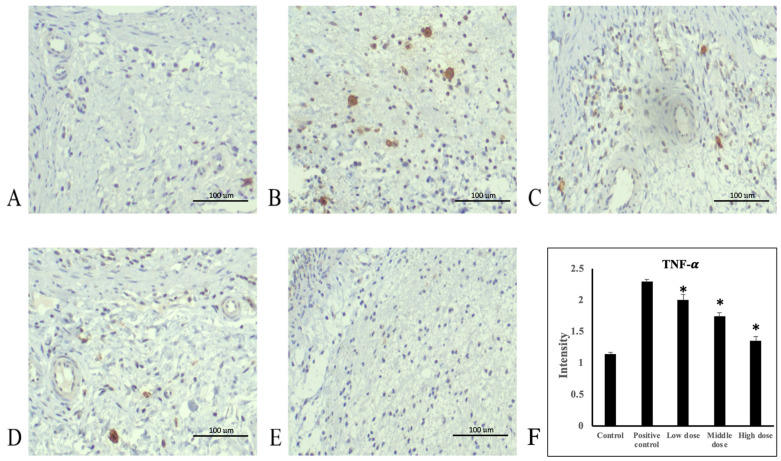
Immunohistochemical staining of TNF-α in stomach tissue. (**A**) Control group, (**B**) positive control group, (**C**) 207.5 mg/kg body weight group, (**D**) 415 mg/kg body weight group, and (**E**) 830 mg/kg body weight group of *A. indica* HP813 powder (100×; brown color represents TNF-α-labeled cells), (**F**) the average intensity of the TNF-α expression by semi-quantitative IHC. *: *p* < 0.05 versus positive control group.

**Figure 5 nutrients-16-02297-f005:**
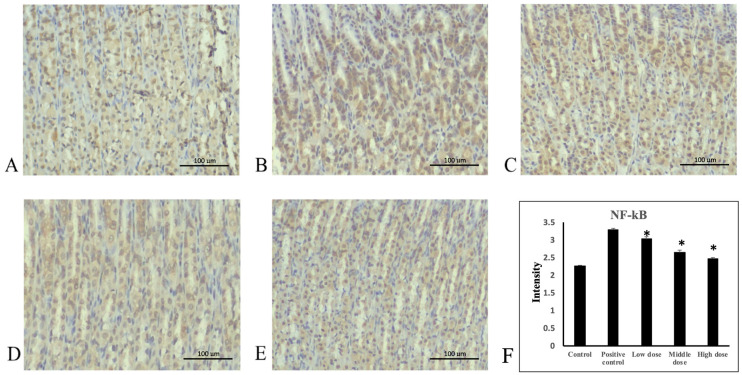
Immunohistochemical staining of NF-κB in stomach tissue. (**A**) Control group, (**B**) positive control group, (**C**) 207.5 mg/kg body weight group, (**D**) 415 mg/kg body weight group, and (**E**) 830 mg/kg body weight group of *A. indica* HP813 powder (100×; brown color represents NF-κB-labeled cells), (**F**) the average intensity of the NF-κB expression by semi-quantitative IHC. *: *p* < 0.05 versus positive control group.

**Figure 6 nutrients-16-02297-f006:**
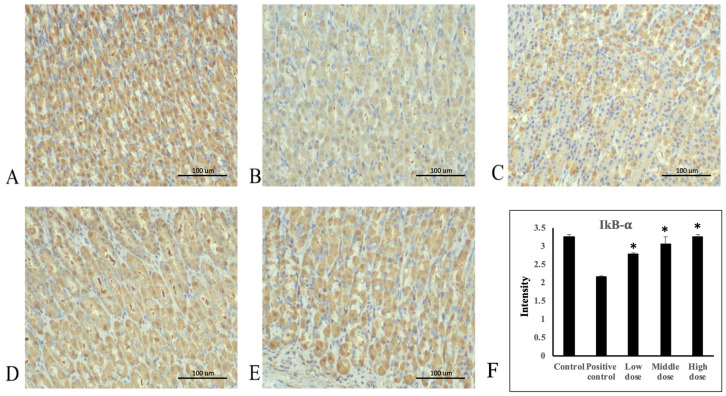
Immunohistochemical staining of IκB-α in stomach tissue. (**A**) Control group, (**B**) positive control group, (**C**) 207.5 mg/kg body weight group, (**D**) 415 mg/kg body weight group, and (**E**) 830 mg/kg body weight group of *A. indica* HP813 powder (100×; brown color represents IκB-α-labeled cells), (**F**) the average intensity of the IκB-α expression by semi-quantitative IHC. *: *p* < 0.05 versus positive control group.

**Table 1 nutrients-16-02297-t001:** The gastric mucosa injury area of rats treated with *A. indica* HP813 powder in the efficacy study.

Group	Total Gastric	Gastric Mucosa Injured Area	
	(cm^2^)	(cm^2^)	(%)
Control	10.03 ± 2.00	0	0
Positive control	11.45 ± 0.82	5.62 ± 2.04 *	48.61 ± 16.24 *
207.5 mg/kg	10.85 ± 0.79	4.90 ± 1.48 *	45.22 ± 13.06 *
415 mg/kg	10.21 ± 0.86	3.64 ± 1.00 *^,a^	35.30 ± 6.97 *^,a^
830 mg/kg	11.09 ± 0.77	3.03 ± 1.10 *^,a^	27.16 ± 8.89 *^,a^

Data are expressed as the mean ± standard deviation (n = 8). * Significant differences among the control, positive control, and treatment groups at *p* < 0.05; ^a^ significant difference between the positive control and treatment groups at *p* < 0.05.

**Table 2 nutrients-16-02297-t002:** A summary of pathological incidence and lesion scores for the protective effects of *A. indica* HP813 powder on the gastric mucosa in rats.

Inflammatory Scores	Group				
	Control	Positive Control	*A. indica* HP813 Powder (mg/kg)		
			207.5	415	830
Stomach					
Degeneration/necrosis, mucosal, multifocal	0.0 ± 0.0	3.1 ± 0.7 *	3.1 ± 0.6 *	2.4 ± 0.7 *	2.3 ± 0.7 *^,a^
Hemorrhage, mucosal, multifocal	0.0 ± 0.0	2.2 ± 0.4 *	2.2 ± 0.4 *	1.8 ± 0.4 *^,a^	1.9 ± 0.3 *
Inflammation, multifocal	0.0 ± 0.0	2.6 ± 0.5 *	2.7 ± 0.5 *	2.6 ± 0.5 *	2.4 ± 1.0 *
Edema, submucosa, multifocal	0.0 ± 0.0	3.9 ± 1.0 *	3.8 ± 0.8 *	2.9 ± 0.7 *^,a^	2.4 ± 1.0 *^,a^
Subtotal lesion score	0.0 ± 0.0	2.9 ± 0.9 *	2.9 ± 0.8 *	2.4 ± 0.7 *^,a^	2.3 ± 0.8 *^,a^

* Statistically significant difference between the control and treatment groups at *p* < 0.05. ^a^ Statistically significant difference between the positive control and treatment groups at *p* < 0.05.

## Data Availability

The original contributions presented in the study are included in the article/Appendix A, further inquiries can be directed to the corresponding author.

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
