# Peer review of "The Gastroprotective Effects of Anisomeles indica against Ethanol-Induced Gastric Ulcer through the Induction of IκB-α and the Inhibition of NF-κB Expression"

_nutrients, 2024, doi:10.3390/nu16142297_

Round 1

Reviewer 1 Report

Comments and Suggestions for Authors

The study is exciting and depicts the potential of the Gastroprotective effects of Anisomeles indica. Any natural compounds/extracts showing functional properties are worthy of the development of functional food products. The MS required revision in the context of the raised queries:

1. complete the statement “I addition, the lesion ……2.3, respectively” in response to what?

2. Rewrite the keywords [Anisomeles indica HP813 powder; ethanol-induced gastric ulcer; NF-kB; IkB-a], which are merely replicas of the title.

3. Place the HO-1 inside the bracket (line 52).

4. Adequately define the study's objective, which has yet to be included in the MS.

4. Line 129, add the Image J software version and date of access.

5. What are the selection criteria for the doses 207.5 mg/kg-830 mg/kg body weight group? Please mention this in the manuscript.  

6. In Figure 4-6, incorporate the scale bar.

7. In image 5 author depicts the gastric injury-including degeneration/ necrosis, hemorrhage, inflammatory response, and edema and later quantified it (Table 2). Please mention these changes using arrows or with different marks to localize them in the tissue section. If this representation is not possible due to the small image size in the MS, the author can provide the enlarged images with the marked changes as a supplementary figure.

8. The graphs derived from Image J or other quantitative software must be added to document the immunobiological quantification depicted in Figures 4-6.

9. Place a “0” before the decimal (p<.05) Line 177, 190; check the other typo errors throughout the MS.

Comments on the Quality of English Language

Minor english editing required.

Author Response

The study is exciting and depicts the potential of the Gastroprotective effects of Anisomeles indica. Any natural compounds/extracts showing functional properties are worthy of the development of functional food products. The MS required revision in the context of the raised queries:

Response: Thank you very much for your professional comments.

  1. complete the statement “I addition, the lesion ……2.3, respectively” in response to what?

Response: We have rewritten the sentence as “In addition, the lesion scores were 2.9, 2.4, and 2.3 in the ethanol-induced group, 415 mg/kg of A. indica HP813 powder group, and 830 mg/kg A. indica HP813 powder group, respectively.” in line 27-28.

  1. Rewrite the keywords [Anisomeles indicaHP813 powder; ethanol-induced gastric ulcer; NF-kB; IkB-a], which are merely replicas of the title. 

Response: We have written the keywords as “A. indica; gastric ulcer; NF-kB; IkB-a” in line 34.

  1. Place the HO-1 inside the bracket (line 52).

Response: We have placed the HO-1 inside the bracket as “(HO-1)” in line 52.

  1. Adequately define the study's objective, which has yet to be included in the MS.

Response: We have added adequately define the study’s objective as: The objective of this study is to determine whether A. indica HP813 powder inhibits ethanol-induced gastric ulcers and whether it can reduce the inflammatory response of gastric ulcers caused by ethanol.” in line 67-69.

  1. Line 129, add the Image J software version and date of access.

Response: We added the ImageJ software version and date of access as “(Version 1.53k, National Institutes of Health, USA. 6 July 2021)” in line 140.

  1. What are the selection criteria for the doses 207.5 mg/kg-830 mg/kg body weight group? Please mention this in the manuscript.

Response: We have added the selection criteria for the doses as “The recommended daily intake of A. indica HP813 powder for an adult weighing 60 kg is 4000 mg, which is 67 mg/kg/day. In terms of experimental animals, multiplying by 6.2 equals the recommended dose of 415 mg/kg/day, 830 mg/kg is twice the recommended dose, and 207.5 mg/kg is half the recommended dose.”, and added in line 94-97.

  1. In Figure 4-6, incorporate the scale bar.

Response: We have added the scale bar in Figure 4A, 4B, 4C, 4D, 4E, Figure 5A, 5B, 5C, 5D, 5E, Figure 6A, 6B, 6C, 6D, and 6E.

  1. In image 5 author depicts the gastric injury-including degeneration/ necrosis, hemorrhage, inflammatory response, and edema and later quantified it (Table 2). Please mention these changes using arrows or with different marks to localize them in the tissue section. If this representation is not possible due to the small image size in the MS, the author can provide the enlarged images with the marked changes as a supplementary figure.

Response: We have provided the enlarged images with the marked changes as a supplementary figure S1A and S1B. “Supplementary Figures S1A and S1B showed gastric injury-including degeneration/ necrosis, hemorrhage, inflammatory response, and edema. The normal control and positive control are shown in Figure S1A, low dose group (207.5 mg/kg), medium dose group (415 mg/kg), and high dose group (830 mg/kg) are shown in Figure S1B.” The sentence is also shown in line 195-199.

  1. The graphs derived from Image J or other quantitative software must be added to document the immunobiological quantification depicted in Figures 4-6.

Response: We have added Figure 4F, Figure 5F, and 6F as the average intensity of the immune reaction by semi-quantitatively IHC, also described “the average intensity of the TNF-α expression by semi-quantitatively IHC were 1.14 ± 0.03, 2.3 ± 0.03, 2.0 ± 0.09, 1.74 ± 0.06, and 1.35 ± 0.07, respectively (Figure 4F, Supplementary Table S1).” in line 217-219, “the average intensity of the NF-κB expression by semi-quantitatively IHC were 2.27 ± 0.01, 3.29 ± 0.04, 3.02 ± 0.07, 2.64 ± 0.07, and 2.48 ± 0.02, respectively (Figure 5F, Supplementary Table S1).” in line 223-225, and “the average intensity of the IκB-α expression by semi-quantitatively IHC were 3.26 ± 0.06, 2.17 ± 0.02, 2.78 ± 0.04, 3.07 ± 0.19, and 3.27 ± 0.05, respectively (Figure 6F, Supplementary Table S1). ” in line 229-231. Table S1 has been uploaded and is shown below:

Table S1. The IHC intensity of immune reaction by semi-quantitatively analyze.

Intensity

TNF-a

NF-kB

IkB-a

Control

1.14 ± 0.03

2.27 ± 0.01

3.26 ± 0.06

Positive control

2.30 ± 0.03

3.29 ± 0.04

2.17 ± 0.02

HP813 Low dose

2.00 ± 0.09*

3.02 ± 0.07*

2.78 ± 0.04*

HP813 Medium dose

1.74 ± 0.06*

2.64 ± 0.07*

3.07 ± 0.19*

HP813 High dose

1.35 ± 0.07*

2.48 ± 0.02*

3.27 ± 0.05*

* Statistically significant difference between the positive control and treatment groups at p < 0.05.

  1. Place a “0” before the decimal (p<.05) Line 177, 190; check the other typo errors throughout the MS.

Response: We placed a “0” before the decimal (p<.05) as p< 0.05 in line 186, 187, 194, 210, and 211.

Reviewer 2 Report

Comments and Suggestions for Authors

Overall, a good manuscript is submitted. Methods are adequately described, but immunohistochemical evaluation has to be repeated. English is acceptable. Figures and tables are of good quality. Some changes in botanical aspects are necessary: Anisomeles indica (L.) Kuntze: the species name has to be written at least once with the author's name. Use Lamiaceae instead of the old family name Labiatae. An explanation on the choice of powder doses has to be added: are these determined due to acteoside content, or some other decision was made? The powder preparation is linked to a previously published paper, but the phytochemical analysis concerns a plant extract, not the powder. So, this manuscript does not provide information on how much acteoside the powder contains. To prove the effect of powder on rats gastric mucosa by immunohistochemical analyses, the Immunohistochemical evaluation has to be determined once again, for example semi-quantitatively in 50 cells of each sample using the following score for cytosol staining: 1 – no, 2 – weak, 3 – moderate, 4 – strong. The average intensity of the immune reaction has to be given as the number of cells of each type x corresponding coefficient (1, 2, 3 or 4) x total number of cells-1.

After such revision, the results must be resubmitted and the discussion updated.

Minor technical changes are needed as well (pdf is uploaded).

Author Response

Overall, a good manuscript is submitted. Methods are adequately described, but immunohistochemical evaluation has to be repeated. English is acceptable. Figures and tables are of good quality.

Response: Thank you very much for your professional comments.

Some changes in botanical aspects are necessary: Anisomeles indica (L.) Kuntze: the species name has to be written at least once with the author's name. Use Lamiaceae instead of the old family name Labiatae. 

Response: We have rewritten “Anisomeles indica (L.) Kuntze” in line 17 and 54. We have used “Lamiaceae” instead of the old family name Labiatae in line 54. 

An explanation on the choice of powder doses has to be added: are these determined due to acteoside content, or some other decision was made?

Response: We have added the selection criteria for the doses as “The recommended daily intake of A. indica HP813 powder for an adult weighing 60 kg is 4000 mg, which is 67 mg/kg/day. In terms of experimental animals, multiplying by 6.2 equals the recommended dose of 415 mg/kg/day, 830 mg/kg is twice the recommended dose, and 207.5 mg/kg is half the recommended dose.” And added in line 94-97.

The powder preparation is linked to a previously published paper, but the phytochemical analysis concerns a plant extract, not the powder. So, this manuscript does not provide information on how much acteoside the powder contains. 

Response: We have added the describe of how much acteoside the A. indica HP813 powder content as: The main efficacy ingredient of A. indica HP813 powder is acteoside. The content of acteoside in A. indica HP813 powder (2000 mg) is 55 mg. In this study, the content of acteoside in the low-dose group was 27.5 mg, the medium-dose group was 55 mg, and the high-dose group was 110 mg.”, and showed in line 77-81.

To prove the effect of powder on rats gastric mucosa by immunohistochemical analyses, the Immunohistochemical evaluation has to be determined once again, for example semi-quantitatively in 50 cells of each sample using the following score for cytosol staining: 1 – no, 2 – weak, 3 – moderate, 4 – strong. The average intensity of the immune reaction has to be given as the number of cells of each type x corresponding coefficient (1, 2, 3 or 4) x total number of cells-1.

Response: According to your suggestion, we used semi-quantitatively to evaluate the immunohistochemical analyses on gastric mucosa and showed as supplementary Table S1. We described as “The immunohistochemistry evaluation was determined by semi-quantitatively in 50 cells of each sample using the following score for cytosol staining: 1 – no, 2 – weak, 3 – moderate, 4 – strong. The average intensity of the immune reaction has to be given as the number of cells of each type x corresponding coefficient (1, 2, 3, or 4) / total number of cells.”, in line 156-160. Results also showed in Figure 4F, 5F, and 6F, also described as “the average intensity of the TNF-α expression by semi-quantitatively IHC were 1.14 ± 0.03, 2.3 ± 0.03, 2.0 ± 0.09, 1.74 ± 0.06, and 1.35 ± 0.07, respectively (Figure 4F, Supplementary Table S1).” in line 217-219, “the average intensity of the NF-κB expression by semi-quantitatively IHC were 2.27 ± 0.01, 3.29 ± 0.04, 3.02 ± 0.07, 2.64 ± 0.07, and 2.48 ± 0.02, respectively (Figure 5F, Supplementary Table S1).” in line 223-225, and “the average intensity of the IκB-α expression by semi-quantitatively IHC were 3.26 ± 0.06, 2.17 ± 0.02, 2.78 ± 0.04, 3.07 ± 0.19, and 3.27 ± 0.05, respectively (Figure 6F, Supplementary Table S1). ” in line 229-231. Table S1 has been uploaded and is shown below:

Table S1. The IHC intensity of immune reaction by semi-quantitatively analyze.

Intensity

TNF-a

NF-kB

IkB-a

Control

1.14 ± 0.03

2.27 ± 0.01

3.26 ± 0.06

Positive control

2.30 ± 0.03

3.29 ± 0.04

2.17 ± 0.02

HP813 Low dose

2.00 ± 0.09*

3.02 ± 0.07*

2.78 ± 0.04*

HP813 Medium dose

1.74 ± 0.06*

2.64 ± 0.07*

3.07 ± 0.19*

HP813 High dose

1.35 ± 0.07*

2.48 ± 0.02*

3.27 ± 0.05*

* Statistically significant difference between the positive control and treatment groups at p < 0.05.

After such revision, the results must be resubmitted and the discussion updated.

Response: We have rewritten the results (3.3, 3.4 use red font) and discussion (use red font) in our manuscript.

Minor technical changes are needed as well (pdf is uploaded).

Response: According to your suggestion, we have revised our manuscript.

Round 2

Reviewer 1 Report

Comments and Suggestions for Authors

The raised query has been resolved and included in the revised manuscript. I recommend proceeding with the publication of the article.

Comments on the Quality of English Language

NA